# Experiences and meanings of integration of TCAM (Traditional, Complementary and Alternative Medical) providers in three Indian states: results from a cross-sectional, qualitative implementation research study

D Nambiar,[1] V V Narayan,[2] L K Josyula,[3] J D H Porter,[4] T N Sathyanarayana,[3] K Sheikh[1]

▶ Prepublication history and additional material is available. To view please visit the journal (http://dx.doi.org/10.1136/bmjopen-2014-005203).

For numbered affiliations see end of article.

**Correspondene to**
Dr Devaki Nambiar;
devaki.nambiar@phfi.org

## ABSTRACT

**Objectives:** Efforts to engage Traditional, Complementary and Alternative Medical (TCAM) practitioners in the public health workforce have growing relevance for India's path to universal health coverage. We used an action-centred framework to understand how policy prescriptions related to integration were being implemented in three distinct Indian states.

**Setting:** Health departments and district-level primary care facilities in the states of Kerala, Meghalaya and Delhi.

**Participants:** In each state, two or three districts were chosen that represented a variation in accessibility and distribution across TCAM providers (eg, small or large proportions of local health practitioners, Homoeopaths, Ayurvedic and/or Unani practitioners). Per district, two blocks or geographical units were selected. TCAM and allopathic practitioners, administrators and representatives of the community at the district and state levels were chosen based on publicly available records from state and municipal authorities. A total of 196 interviews were carried out: 74 in Kerala, and 61 each in Delhi and Meghalaya.

**Primary and secondary outcome measures:** We sought to understand experiences and meanings associated with integration across stakeholders, as well as barriers and facilitators to implementing policies related to integration of Traditional, Complementary and Alternative (TCA) providers at the systems level.

**Results:** We found that individual and interpersonal attributes tended to facilitate integration, while system features and processes tended to hinder it. Collegiality, recognition of stature, as well as exercise of individual personal initiative among TCA practitioners and of personal experience of TCAM among allopaths enabled integration. The system, on the other hand, was characterised by the fragmentation of jurisdiction and facilities, intersystem isolation, lack of trust in and awareness of TCA systems, and inadequate infrastructure and resources for TCA service delivery.

## Strengths and limitations of this study

- Multisited qualitative study drawing on meanings and experiences across patients, providers and health systems administrators.
- Implementation research using rigorously applied interpretive policy analysis methods.
- Linked to India's path to Universal Health Coverage.
- Cross-sectional study, so other than self-report of historical changes, we were not able to chart or map changed views or experiences of participants in vivo.
- Focus on the public service delivery sector, even as a great deal of health seeking takes place in the private sector, with the assumption that public sector strengthening is highly desirable and possible only through a focused study on it.

**Conclusions:** State-tailored strategies that routinise interaction, reward individual and system-level individual integrative efforts, and are fostered by high-level political will are recommended.

## INTRODUCTION

The 1978 Alma Ata declaration called for traditional medicine treatments and practices to be "preserved, promoted and communicated widely and appropriately based on the circumstances in each country." Thirty years later, the 2008 Beijing Declaration on Traditional Medicine called for the integration of providers into national health systems, recommending systems of qualification, accreditation, regulation and communication (with allopathic providers).[1] These features of the Beijing Declaration were echoed at the

62nd World Health Assembly in 2009, putting out a call to action to United Nations member states to move forward with their plans for integration.[2] The global positioning of Traditional, Complementary and Alternative Medicine (TCAM) has issued from and tends to imply a central focus on clinical and experimental medicine,[3] yet recent calls for health systems integration draw attention to features such as education, accreditation, regulation and health services provision, and the TCAM health workforce itself.

In an earlier study, we have identified three broad trends of integration as it relates to TCA providers: self-regulation with governmental linkage, government regulation and provisioning, and hybrid/parallel models.[4] This links roughly to the WHO nosology, where three models are identified: 'tolerant' systems, where the national healthcare system is based entirely on biomedicine but some TCAM practices are legally permissible; 'inclusive' systems, where TCAM is recognised but not fully integrated into all aspects of healthcare; and 'integrative systems,' where TCAM is officially recognised in national drug policy, providers and products are registered and regulated, therapies are widely available and covered under insurance schemes, and research and education are widely accessible.[5]

The situation on the ground in India, hybrid in our view, seems in parts to reflect tendencies across the WHO categories. The dominance of biomedicine appears to be a critical feature of India's postcolonial health system, even as pre-independence the TCAM practitioner community had played a major role in resisting colonial domination in the practice of (bio)medicine.[6] In part as a response to the reliance on allopathy throughout modern Indian history, there have been strong arguments in favour of the critical role that non-mainstream practitioners play in offering accessible, affordable and socially acceptable health services to populations.[1 7 8] A study in Maharashtra reported that the situation of traditional healing as a community function through shared explanatory frameworks across provider and patient is explicitly unlike typical doctor–patient relationships.[9]

In India, one can also find a larger integrative framework, one that mandates the 'mainstreaming' of codified TCAM in India, collectively referred to as AYUSH, an acronym for Ayurveda, Yoga and Naturopathy, Unani, Siddha, Sowa-Rigpa and Homoeopathy. The National Rural Health Mission (NRHM), launched in 2005 to fortify public health in rural India, took particular interest in integrating AYUSH practitioners through facilitation of specialised AYUSH practice, integration of AYUSH practitioners in national health programmes, incorporation of AYUSH modalities in primary healthcare, strengthening the governance of AYUSH practice, support for AYUSH education, establishment of laboratories and research facilities for AYUSH, and providing infrastructural support.[10] Human resource-focused strategies included the contractual appointment of AYUSH

doctors in Community and Primary Health Centres (PHCs), appointment of paramedics, compounders, data assistants and managers to support AYUSH practice; establishment of specialised therapy centres for AYUSH providers; inclusion of AYUSH doctors in national disease control programmes; and incorporation of AYUSH drugs into community health workers' primary healthcare kits. A recent report from the AYUSH department states that NRHM has established AYUSH facilities in co-location with health facilities in many Indian states (most notably not in Kerala, where the stand-alone AYUSH facility is the chosen norm).[11] As of 2012, more than three quarters of India's district hospitals, over half of its Community Health Centres and over a third of India's PHCs have AYUSH co-location, serving about 1.77 million, 3.3 million and 100 000 rural Indians, respectively.[11]

Yet even this integration framework has at most an 'inclusive' character. This is reflected in findings such as 'official neglect' of traditional orthopaedic practitioners who have no registration, uniformity in interstate regulation, or institutionalised medical training.[12] AYUSH doctors contracted to Medical Officer posts in PHCs in the southern Indian state of Andhra Pradesh report numerous lacunae in the implementation of the mainstreaming initiatives in the NRHM:[13] job perquisites are not indicated; no benefits or allowances are provided for health, housing or education, and compensation packages are much lower than those of allopathic doctors. Support for AYUSH practice is also inadequate (lack of infrastructure, trained assistants and drug supply) and unethical practices have also been reported (documenting attendance of absentees, and non-cooperation from non-AYUSH personnel). Evidence from NRHM suggests that reshuffled AYUSH providers practise forms of medicine beyond the scope of their training.[14] Paradoxically, moreover, some Indian states prohibit cross-system prescription, adding ethical dilemmas for TCA practitioners who serve as the only medical practitioners in resource-poor areas.[14]

On a larger scale, current practices of integration (as in NRHM) have been described as substitution and replacement; which tend to ignore the merits of TCAM and present more barriers than facilitators of integration.[7] In particular, given the strong push towards co-location and other strategies of integration as part of India's move towards Universal Health Coverage, the integration of AYUSH practitioners could result in a doubling of the health workforce. Yet there are strong fears that such an emphasis on quantitative aspects of integration, that is, having the right number of practitioners placed at facilities, is inadequate. There is a need to critically and qualitatively appraise the government infrastructure to support TCA, identify barriers and facilitators to integration that have emerged from this rapid placement of these practitioners, and how these TCA practitioners, allopathic practitioners and health system actors are reacting and adapting to each factor.

## METHODS

This analysis draws from a larger mixed-methods implementation research study aimed at understanding operational and ethical challenges in integration of TCA providers for delivery of essential health services in three Indian states. The study looked at the contents and implementation of TCA provider integration policies in three states, and at the national level it examined the understanding and interpretations of integration from the perspectives of different health system actors. These, coupled with their experiences in the actual processes of integration of TCA providers, were studied using qualitative interview methods to help identify systemic and ethical challenges. Based on this, the study sought to derive strategies to augment the integration of TCA providers in the delivery of essential health services.

Our study was based on action-centred frameworks[15] with a focus on policy *actors* and *processes*.[16] We have therefore sought to understand the implementation of integration policies empirically. A team of four field researchers was oriented by the principal investigator and advisor to the postpositivist paradigm of research, using Yanow's model of interpretative policy analysis, where the emphasis is equally on describing the experience of policy processes, and on elaborating the meanings actors attach to those processes.[17]

Our methods included semistructured in-depth interviews (see interview guides, online supplementary appendix 1) with policymakers (N=12), administrators (N=43), TCAM practitioners (N=59), allopathic practitioners (N=37), traditional healers (N=7), as well as health workers and community representatives (N=38) in three diverse Indian states (figure 1). We undertook the study in Kerala, where a number of systems have strong historical and systemic roots (N=74); Meghalaya, where local health traditions hold sway (N=61); and Delhi, where national, state and municipal jurisdictions interface with multiple systems of medicine (N=61). Participants were selected based on maximum variation criteria for each category. We sought to represent different schemes, levels of implementation (directorates, zonal officers), systems of medicine, types of establishments (hospital, dispensary) and years of experience.

In each state, one senior researcher, a research associate and a field researcher developed selection matrices to achieve maximum variation across each category of respondents. In each state, districts (two in Kerala, and three in Meghalaya) or municipal zones (three in Delhi) were chosen to represent variation in accessibility and distribution across TCA providers (eg, small or large proportions of local health practitioners, Homoeopaths, Ayurvedic and/or Unani practitioners). Publicly available records from state and municipal authorities were consulted in order to determine the location and type of facility (co-located, stand-alone) as well as suggestions and recommendations from key informants. We also ensured that facilities closest to and furthest from district headquarters were chosen for interviews, to maximise variability. We would typically contact providers via cell phone, share information about the study verbally or via email, and set up a time to interview them in-person. In some cases, we would arrive during outpatient clinic hours at the chosen facility, share our participant information sheet and seek an appointment time with eligible participants. In most cases, we found that participants were keen to participate once they were aware of the nature of the study and, in some cases, the assurance of confidentiality. We had no refusals, although some allopathic practitioners had to be persuaded to participate by emphasising that this study was not 'pro-TCAM integration' per se, but merely seeking to understand state policy implementation. Interviews, ranging from 15 to 90 min in length, were undertaken, always with prior informed consent, and with separate consent to record interviews. Data were transcribed and stored in password-protected folders and each transcript was checked by investigators for accuracy and quality of transcription.

Textual data from transcripts of interviews as well as notes and observations of facilities and service delivery recorded during fieldwork were analysed through a combination of deductive and inductive techniques in the 'framework' approach of qualitative analysis for applied policy research[18] using ATLAS.ti7 software. Themes were developed in three iterations: in the first stage, the lead researcher from each state applied a priori codes and closely perused transcripts to devise *emergent* codes, with the support of the Research Associate. The a priori codes were based on our research questions, reflecting experiences, interpretations and meanings of integration. *Emergent* codes were used to describe the content or categories of these experiences, interpretations and meanings. Researchers coded 20% of each other's state data sets to ensure that codes were being applied in a similar, uniform manner. In the second stage, agreement and consolidation of emergent codes across three sites took place under the direction of the study lead; these were then applied to data from each state by its respective lead researcher. Concurrently, lead researchers developed super codes, or *analytic* codes, to group emergent codes. The study lead finalised and then indexed these codes across sites to arrive at results. Emergent and analytic code families were used to develop analyses, involving sharing of data and consultation across sites. In this paper, we focus on emergent codes related to the experiences and interpretations of integration.

## RESULTS

We found that facilitators of integration emerged from individual and interpersonal relationships, while barriers were identified at the systems level (table 1).

### Facilitators at the individual/interpersonal level
#### Collegiality between practitioners within facilities
Interpersonal collegiality was reported between and across some TCA and allopathic practitioners. In

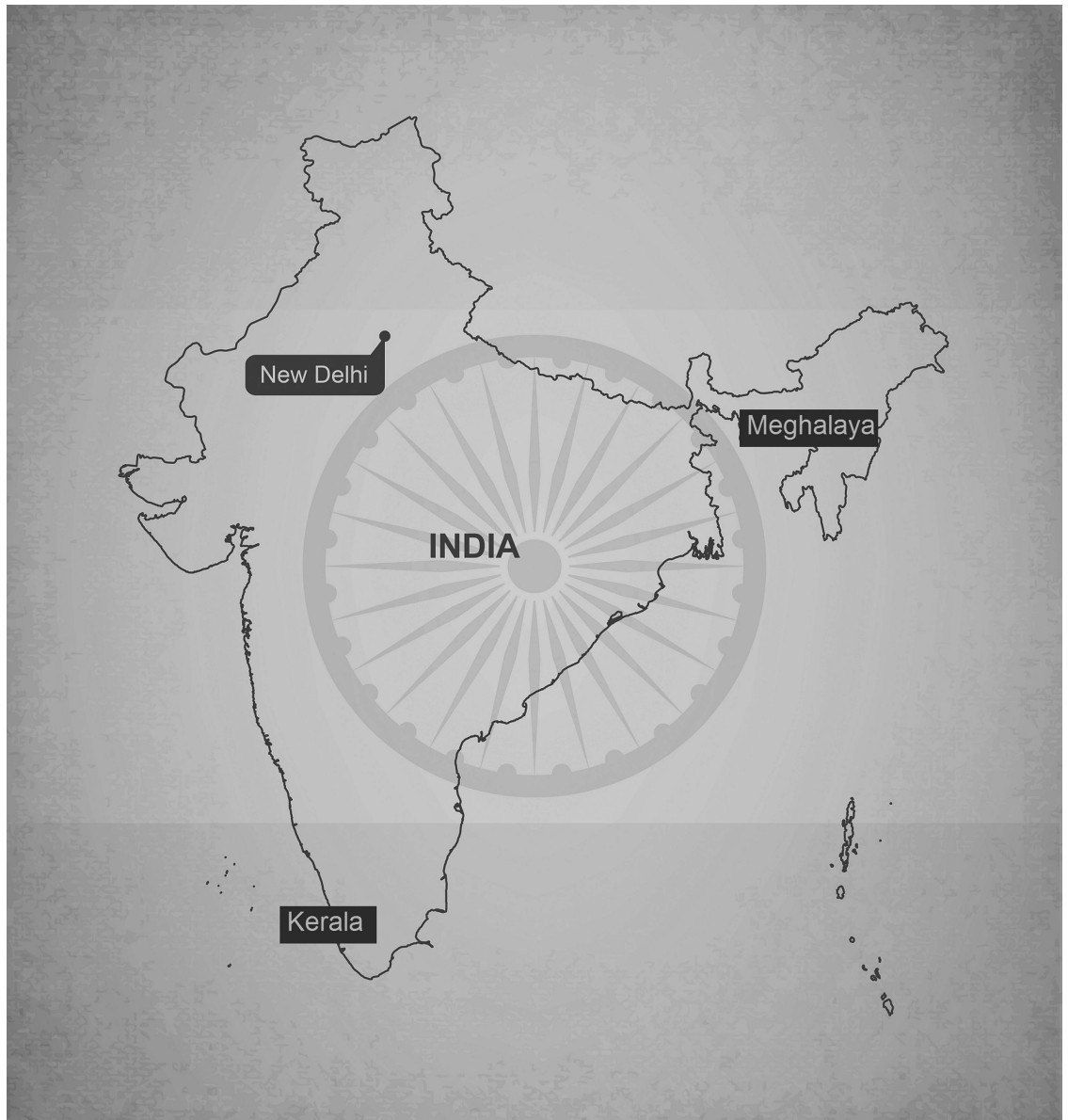

**Figure 1**  Location of states in India where fieldwork was conducted (New Delhi, Meghalaya and Kerala).

Meghalaya, an allopathic medical officer noted that in some places Ayurvedic and Homoeopathic doctors were collaborating closely with their allopathic colleagues, expressing an interest in learning more about allopathic practices. In the same state, an AYUSH doctor described cordial relations with the administration, such that when medicine stockouts happened, the allopathic medical officer supplied stopgap funds to acquire medicines.

### Stature of TCA doctors

Another aspect was the 'stature' of individual practitioners. In Kerala, an Ayurvedic practitioner noted that: "Nobody can question <Name of Well Known Ayurvedic Physician from Kerala>. If he says that taking *chavanaprasham* (health paste) will lead to DNA repair, then nobody can question because they are saying with authority. They are beyond questioning. If somebody else is saying (the same thing), they will ask, where is the proof?" This was also the case with a private sector entity that had opened a branch in Delhi. Practitioners in this institution were highly reputed, involved with transnational research collaborations, and reported numerous cross-referrals from allopathic providers across the city.

### Personal initiative of TCA doctors

Across states, we heard of individual TCA practitioners exercising personal initiative to hasten improvements in infrastructure and service delivery. The following is an excerpt of an interview with an Ayurvedic doctor from a Delhi hospital: "There is a lack of storage space so the diagnosis room is being used for some storage. But I

| Table 1 | Summary of findings | |
|---|---|
| **Factors at the individual/interpersonal level** | **Factors at the group/system level** |
| *Facilitators* | *Barriers* |
| A. *Collegiality* between practitioners within facilities | A. *Fragmentation* of jurisdiction and facilities |
| B. *Stature* of TCA doctors | B. Intersystem *isolation* and lack of communication |
| C. *Personal initiative* of TCA doctors | C. Lack of *trust and awareness* of TCA systems |
| D. *Personal experience* of allopaths | D. *Inadequate infrastructure and resources* for TCA service delivery |
| E. *Political will* of senior health system actors | |

have been treating people in the Public Works Department and then it is getting resolved!" Many of the participants we spoke to in many states were familiar with each other—these personal relationships and interactions, in the absence of official or regular platforms, were the basis for interaction, cross-referral, collective planning and advocacy, and, in rarer cases, collaborative research.

### Personal experience of allopaths

Personal experience across systems also helped build trust. In Kerala, an allopath indicated that his own mother-in-law was under Ayurvedic treatment for chronic illness and that she and others he knew were "getting good relief." He noted that Ayurveda was trustworthy based on this experience. As an Ayurvedic practitioner in Delhi put it, "if one takes a personal interest, there can be a little something. But everyone is busy in their own work. If it is done officially—like in a month, every 2nd Saturday… Then it will happen more systematically."

### Political will of senior health system actors

Systems level integration was facilitated by highly networked individuals and/or individual access to top decision-makers. One of the health system actors we interviewed had participated in high-level negotiations with political leaders in the country to get the AYUSH department formed (formerly the Indian Systems of Medicine and Homoeopathy department) in 1995—which in many ways marks a critical step in the attention given to integration in the health system. Within the state of Delhi, furthermore, it was the demand articulated by city councillors and ward leaders that resulted in the construction of dispensaries and AYUSH wards in hospitals, so much so that this was considered a norm.

### Barriers at the systems level
### Fragmentation of jurisdictions and facilities

It was clear that systematic integration was not widely perceived in any of the facilities or states studied. For one, all states did not have a single unified system; rather, there existed multiple systems with parallel governance apparatuses, each with their own challenges. In fact, in Delhi, integration was constrained in the system not only by the fragmentation of jurisdictions and

facilities, but also with respect to how providers were posted at facilities. In this state, co-location did take place, but it involved an individual TCA practitioner being co-located at multiple sites, while multiple allopaths served at a single site (the biomedical norm). Allopaths had more opportunities, in terms of sheer numbers of people and availability of space and time, to communicate with each other.

### Intersystem isolation and lack of communication

Given the aforementioned lack of people, space and time, allopaths were socially isolated from and had fewer chances to communicate with TCA providers, or TCA providers with each other.

In Kerala, the limitations on communication were shaped in particular by the fact that facilities tended to be stand-alone. In Meghalaya, an allopath stated simply, "I am doing my work, and they (TCA providers) are doing theirs… that is completely asocial type, separated, segregated." There was almost no communication between local health practitioners and others—whether AYUSH or allopath—simply because of a lack of systemic acknowledgement and legitimacy given to this workforce. A TCA provider remarked, "Very few people listen to our problem. Because, we are still, again, you know, under the general allopathic doctor,… so when we post our problem you know, hardly like, they table that problem…"

### Lack of trust and awareness of TCA systems

When speaking about providers as a cadre, group or systems in general, we noted that distrust tended to be highlighted. In Meghalaya, an allopath opined, "Please, if you want us to work in a normal way, you know, peacefully, just have these people removed." A similar sentiment was expressed by a senior Unani hospital practitioner in Delhi, "We can interact as a *pathy* but our basic concepts do not match. We can't help each other in any way. They are independent, we are independent." There was limited value, in the view of this practitioner, in engaging with other systems of medicine. An allopath in Kerala described at length how allopathic doctors had protested vehemently—and successfully—against a government policy of Ayurveda doctors getting house surgeon postings in the state. More junior practitioners noted that even with respect to TCAM systems: "We three (Ayurveda, Unani and Homoeopathy) are

**Table 2** Strategies to increase facilitators and decrease barriers to integration, corresponding with the study findings

| Strategies that may enhance TCA integration for essential health services delivery, based on our findings | Strategies that promote Facilitators | | | | Strategies that remove Barriers | | | |
|---|---|---|---|---|---|---|---|---|
| | Collegiality | Stature | Personal initiative | Personal experience | Fragmentation | Isolation | Lack of trust/ awareness | Inadequate infrastructure/ resources |
| High-level political will required for all strategies | | | | | | | | |
| Case documentation and sharing across systems, and in the academic literature | | ✓ | ✓ | ✓ | | ✓ | ✓ | |
| Routine opportunities for interaction and collaboration across systems (eg, health camps, health promotion drives) | ✓ | | | ✓ | ✓ | ✓ | ✓ | ✓ |
| Routine opportunities for interaction within co-located facilities (eg, staff meetings) | ✓ | | | ✓ | ✓ | ✓ | ✓ | ✓ |
| Rewards for the integrative initiative of individuals (eg, challenge grants or institutional recognition) | ✓ | ✓ | ✓ | ✓ | | | | |
| Rewards for the integrative initiative at the systems or facility level (eg, joint targets such as the number of monthly referrals, number of cases resolved jointly) | | ✓ | | ✓ | ✓ | ✓ | ✓ | ✓ |
| Guidelines for collaboration (criteria and conditions for cross-referral, jointly developed by practitioners, non-clinical aspects of work together, including health promotion and managerial duties) | | | | ✓ | ✓ | ✓ | ✓ | ✓ |

together here, but cross-reference is very, very less…We don't know what is the strong point of Ayurveda, Unani. Allopath will not know the strong point of Homoeopathy, Ayurveda. They just say 'skin!'—that's all they know!"

## Inadequate infrastructure and resources for TCA service delivery

Opportunities to interact were further constrained by the system design of service delivery. We observed in many dispensaries and hospitals in Delhi that non-allopathic practitioners were assigned rooms on the top floor of the facility, while allopaths were allocated multiple rooms on the ground floor (fieldnotes 11, 20, 21, 22 and 27 June 2012). Most commonly, the kinds of cases that they were handling included orthopaedic ailments, and other conditions (motor, neurological and gastric) that constrained mobility and created a very real barrier of access to care within a healthcare facility for patients. Practitioners therefore spend much of their time responding to these inadequacies.

There were also shortcomings in the design of diagnostic services and an inadequacy of human resources. Homoeopathic and Ayurvedic practitioners in Kerala noted the recourse to outsourcing diagnostic investigations because of the lack of facilities in their institutions. Further, there was reliance on the contractual recruitment of human resources to address shortages, which, in their view, affected the stability and reliability of service delivery. When we asked an administrator of one of Delhi's newest, state-of-the-art Ayurvedic facilities what kind of coordination occurred across departments as part of the hospital's functioning, he shrugged and replied, 'Nothing as such!'

## DISCUSSION

The most striking feature in our findings is the emergence of individual experiences and interpretations as enablers or facilitators of integration, in the form of collegiality, recognition of stature, exercise of personal initiative among TCA practitioners and of personal experience of TCAM among allopaths. In contrast, barriers to integration seemed to exist at a systems level. They included fragmentation of jurisdiction and facilities, intersystem isolation, lack of trust in and awareness of TCA systems, and inadequate infrastructure and resources for TCA service delivery. It is a system where 'little somethings' of individuals that catalyse integration are met with 'nothing as such' at the systems level.

Some of our findings are not new—the experience of a lack of interaction has emerged in Hollenberg's study on an integrated practice, which reported that weekly doctors' meetings included only biomedical doctors, not CAM.[19] This study also reported the 'geographical dominance' of biomedical doctors in terms of location of consulting rooms, as was found in our study. A study by Broom et al[20] found tension and mistrust, as well as

inconsistencies in practice and values related to biomedicine and TCAM, among Indian oncologists. Such challenges were also seen in our study.

Our study also revealed some unique findings with respect to the extant literature. Chung et al[21] attributed low referrals from biomedicine to TCAM in Hong Kong to the lack of articulated and enforced procedures of referral in an integrated medical establishment. In the Indian case, it appears that the vagueness of process allows ad hoc interactions and referrals based on personal rapport and, at the same time, discourages the kind of predictable, routine interactions that would allow such rapport to be built. Speaking of integration of Sowa-Rigpa in Bhutan since 1967, Wangchuk et al[22] suggest that there are managerial lessons offered by the juxtaposition and collaboration of conceptually distinct systems within a single administrative and policy unit, such as a ministry. In effect, as they point out, services may not be co-located, but their administration necessarily should be. One could argue that India's case is different—whether in facilities or administratively, it is not just two systems, but more like eight (across AYUSH systems), that are to be integrated, introducing internal hierarchies and complexities that are unique to the country.

In the 1990s and early 2000s, it was argued that integration is about a 'battle between two scientific truths,'[23] or that the CAM field creates two tendencies: "uninformed skeptics who don't believe in anything, and uncritical enthusiasts who don't care about data."[24] Analysis of service delivery in India over a decade later suggests that there are multiple battles being fought—epistemological, logistical, ethical and operational across systems, with (re)conciliatory intercession, at times, of individuals.

How can such intercessions be encouraged, even catalysed? We offer a few suggestions for activities in the Indian case that leverage the individual facilitators of integration to fill systemic gaps (table 2). These strategies are based on the aforementioned findings in particular states; their 'translate-ability' to other states would have to be examined.

For one, improved documentation of clinical cases across systems could be undertaken and shared. We noted that those AYUSH practitioners who were documenting their practices had greater stature, opportunities and topics for interaction with peers. Drawing on personal initiative and creating experiences of interaction could help raise the stature of TCA practice while also reducing isolation and the lack of awareness. State health departments could create routine opportunities for interaction and collaboration across systems, and within facilities. In Delhi, polio immunisation has served as an integrative platform for many practitioners to work together and develop trust and ties. Within facilities, joint staff meetings may serve a similar purpose. Authorities may also consider rewarding individual initiatives for integration (through challenge grants or

institutional recognition)—these could be designed to address systems-level barriers to integration. Systems integration could also be rewarded, through joint or synergistically achieved targets for referrals, or the number of patients cared for using complementary or adjuvant therapies. As of now, those reporting cross-referrals only know of each other; if targets were set, there would be greater incentives for and attention to conditions and protocols for cross-referral. Many practitioners we spoke to suggested that guidelines for collaboration (including cross-referral) be created. We feel that this itself could be a starting point of collaboration among TCA providers and with allopathic providers. In each state, the feasibility of each of these strategies would have to be determined, given the due attention through the exertions of powerful stakeholders with a political will who at various points may find themselves battling each other over policies or power.

## CONCLUSION

Battles occur between armies, while acts of diplomacy involve intricate latticework relationships among individuals with overlapping needs and interests. Our research across three very different Indian states—Kerala, Meghalaya and Delhi—suggests that strategies that attempt to make the health systems receptive to individual integrative efforts may facilitate integration across systems, creating opportunities for greater collaboration, and trust. We have proposed strategies to this end, which must in turn be additionally tailored to each state context, so that the health system exists in a vibrant as well as coherent plurality of human agency.

**Author affiliations**
[1]Public Health Foundation of India, New Delhi, India
[2]All India Institute of Medical Sciences, New Delhi, India
[3]Indian Institute of Public Health, Hyderabad, India
[4]Department of Global Health and Development, London School of Hygiene and Tropical Medicine, London, UK

**Acknowledgements** The authors are grateful for the field support of Kaveri Mayra, Candida Thangkhiew, Bobylin Nadon, Darisuk Kharlyngdoh, Ivanhoe Marak and Sabitha Chandran; and for the guidance of Dr Sandra Albert.

**Contributors** KS and JDHP made substantial contributions to the conception or design of the work. DN, VVN, JKL and TNS made substantial contributions to the acquisition of data. All authors contributed substantially to the analysis and interpretation of data for the work. With DN playing a lead, coordinating role in drafting the work, all authors revised it critically for important intellectual content, giving final approval of the version to be published. Further, all authors agree to be accountable for all aspects of the work in ensuring that questions related to the accuracy or integrity of any part of the work are appropriately investigated and resolved.

**Funding** This research was supported by a Wellcome Trust Capacity Strengthening Strategic Award to the Public Health Foundation of India and a consortium of UK universities (grant number WT084754).

**Competing interests** None.

**Ethics approval** Institutional Ethics Committee of the Public Health Foundation of India.

**Provenance and peer review** Not commissioned; externally peer reviewed.

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
