## [Reviewer comments · BMJ Open]

This paper was submitted to the JECH but declined for publication following peer review. The authors addressed the reviewers' comments and submitted the revised paper to BMJ Open. The paper was subsequently accepted for publication at BMJ Open.

ARTICLE DETAILS

TITLE (PROVISIONAL)	Experiences and meanings of integration of TCAM (Traditional, Complementary and Alternative Medical) providers in three Indian states: Results from a cross-sectional, qualitative implementation research study
AUTHORS	Nambiar, Devaki; Narayan, Venkatesh; Lakshmi, JK; Porter, John; Sathyanarayana, TN; Sheikh, Kabir

VERSION 1 - REVIEW

REVIEWER	Eran Ben-Arye Integrative Oncology Program, The Oncology Service and Lin Medical center, Clalit Health Services, Haifa and Western Galilee District, Israel; and Complementary and Traditional Medicine Unit, Department of Family Medicine, Faculty of Medicine, Technion-Israel Institute of Technology, Haifa, Israel
REVIEW RETURNED	21-Apr-2014

GENERAL COMMENTS	This is an important large-scale study performed in the field of integrative medicine. There is some discrepancy between the high standard of writing in the introduction and methods sections compared with the results section. I would recommend focusing the results section by presenting the findings on several theme axis (e.g. integrative vs. alternative conceptualization, patient-centered care vs. disease-oriented approach). In the discussion section, I recommend adding point-to-point recommendations for further integration based on each of the findings indicated in the results section. It is advisable to generalize your recommendations so that other scholars who promote integration worldwide will benefit from your experience. In addition, please add a small map indicating the 3 research areas for those readers who are not acquainted with the geography of India.
--

REVIEWER	Barbara J. Stussman Survey Statistician Office of the Director National Center for Complementary and Alternative Medicine National Institutes of Health U.S.A.
REVIEW RETURNED	24-Apr-2014

GENERAL COMMENTS

Stussman comments on bmjopen-2014-005203

Abstract

Participants/2nd sentence:

Explain, "utilizing criteria of proximity from district headquarters" (i.e. did you use districts close by or did you vary the proximity?) If the former is true, this should be discussed in limitations.

Methods

First paragraph: Is the larger research study published? If so, cite it. If not, provide a brief background on it.

3rd paragraph:

- suggest providing number in parentheses after each category of respondent to show how many interviews were conducted with each.
- State the length or range of length of each interview.
- State mode of interviews (e.g., face-to-face, telephone).

4th paragraph:

- State how respondents were approached and if you experienced hesitance in participating on the part of the respondent. If so, how did you counteract this hesitance?
- Provide the interviewing protocol as an appendix or describe it in the methods section.

5th paragraph:

- 2nd sentence: Provide information about the a priori codes and how they were developed.
- 3rd sentence: what percent of datasets were double coded?

Results

Individual experiences and meanings – collaboration and trust:

- 2nd paragraph/1st sentence: spell out what MSV stands for.
- 3rd paragraph: Use more quotes from respondents or state how the respondents' perspectives are being relayed. This paragraph reads more like background than findings.
- 5th paragraph/3rd and 4th sentences: Elaborate on what is meant by "little something."

Group or system-linked experiences and meanings-distrust and fragmentation:

- 1st paragraph/last sentence: This sentence seems contrary to the section heading. Consider adding a section on differences observed across ages if you have enough data to do so.
- 3rd paragraph/2nd sentence: Rather than describing what you observed, use quotes or summaries of what the respondents said. For example "non-allopathic practitioners talked about the room assignments..." If you want to talk about your own observations, this should be added to the methods section as a data source and the methods of observation should be described.

Conclusion

The conclusion would be more salient if one or two examples were provided of an individually tailored strategy that would aid in integration.

General comment

The manuscript needs copy editing.

VERSION 1 – AUTHOR RESPONSE

-Reviewer(s) Reports:

Reviewer: 1

Reviewer Name Eran Ben-Arye

Institution and Country Integrative Oncology Program, The Oncology Service and Lin Medical center, Clalit Health Services, Haifa and Western Galilee District, Israel; and Complementary and Traditional Medicine Unit, Department of Family Medicine, Faculty of Medicine, Technion-Israel Institute of Technology, Haifa, Israel

Please state any competing interests or state 'None declared': None declared

This is an important large-scale study performed in the field of integrative medicine. There is some discrepancy between the high standard of writing in the introduction and methods sections compared with the results section. I would recommend focusing the results section by presenting the findings on several theme axis (e.g. integrative vs. alternative conceptualization, patient-centered care vs. disease-oriented approach).

The need to present findings on theme axes is well taken. We have done this in the body of the manuscript and also provided a summary table for clarity.

Table 1. Summary of Findings (hard to view in plain text; please see supplementary file)

In the discussion section, I recommend adding point-to-point recommendations for further integration based on each of the findings indicated in the results section. It is advisable to generalize your recommendations so that other scholars who promote integration worldwide will benefit from your experience.

Reviewer 2 asked for state-tailored recommendations, but we tend to agree with this reviewer that the lessons may be relevant in other country contexts. We have presented them in relation to our findings and further argued for their customization in each state.

Table 2. Recommendations to promote/address Integration, responding to findings (hard to see in plain text; please see supplementary file)

In addition, please add a small map indicating the 3 research areas for those readers who are not acquainted with the geography of India.

We have found an open-source map and indicated the three areas as Figure 1. We have reproduced this image below.

Figure 1. States in India where fieldwork was conducted (not visible in plain text; please see supplementary file)

Reviewer: 2

Reviewer Name Barbara J. Stussman

Institution and Country Survey Statistician

Office of the Director

National Center for Complementary and Alternative Medicine

National Institutes of Health
U.S.A.

Please state any competing interests or state 'None declared': None declared

Abstract

Participants/2nd sentence:

Explain, "utilizing criteria of proximity from district headquarters" (i.e. did you use districts close by or did you vary the proximity?) If the former is true, this should be discussed in limitations. Of the two administrative regions chosen within district, the nearest and farthest regions from district headquarters were selected for interviewing the participants. We have indicated this on page 7.

Methods

First paragraph: Is the larger research study published? If so, cite it. If not, provide a brief background on it.

The larger research study is reported, but not published. We have provided a brief background on page 6, as follows. "This analysis draws from a larger mixed methods implementation research study aimed at understanding operational and ethical challenges in integration of TCA providers for delivery of essential health services in three Indian states. The study looked at the contents and implementation of TCA provider integration policies in 3 states and at national level examining the understanding and interpretations of integration from the perspectives of different health systems actors. These coupled with their experiences in the actual processes of integration of TCA providers were studied using qualitative interview methods to help identify systemic and ethical challenges. Based on this, the study sought to derive strategies to augment the integration of TCA providers in the delivery of essential health services."

3rd paragraph:

- suggest providing number in parentheses after each category of respondent to show how many interviews were conducted with each.

This has been indicated as follows on pages 6-7: "Our methods comprised semi-structured, in-depth face-to-face interviews with policymakers (N=12), administrators (N=43), TCAM practitioners, (N=59) and allopathic practitioners (N=37), traditional healers (N=7), as well as health workers and community representatives (N=38) in three diverse Indian states: Kerala, where a number of systems have strong historical and systemic roots (N=74), Meghalaya, where local health traditions hold sway (N=61), and Delhi, where national, state, and municipal jurisdictions interface with multiple systems of medicine (N=61).

- State the length or range of length of each interview.

This has been indicated on page 7 of the paper, as follows: "Interviews, ranging from 20 to 90 minutes in length were undertaken, only with prior informed consent, and separate consent to record interviews."

- State mode of interviews (e.g., face-to-face, telephone).

The modality was face-to-face interviewing, which has been indicated on page 6 (please see quote above)

4th paragraph:

- State how respondents were approached and if you experienced hesitance in participating on the

part of the respondent. If so, how did you counteract this hesitance?

This information has been added into Page 7 as follows: “We would typically contact providers via cell phone, share information about the study verbally or via email, and set up a time to interview them. In some cases, we would arrive during out-patient clinic hours to the chosen facility, share our participant information sheet and seek an appointment time with eligible participants. In most cases, we found that participants were keen to participate once they were aware of the nature of the study and, in some cases, the assurance of confidentiality. We had no refusals, although some allopathic practitioners had to be persuaded to participate by emphasizing that this study was not “pro-TCAM integration” per se, but merely seeking to understand state policy implementation.”

- Provide the interviewing protocol as an appendix or describe it in the methods section.

The interviewing protocol is being submitted as an appendix.

5th paragraph:

- 2nd sentence: Provide information about the a priori codes and how they were developed.

A priori codes were derived directly from our research questions. This is indicated on page 8 as follows: “A priori codes were based on our research questions, reflecting experiences, interpretations and meanings of integration (eg. Tc_Ap_EI_Adm refers to a TCAM providers’ explanation of experience of interactions with administration in the facility or the health care system). Emergent codes were used to describe the content or categories of these experiences, interpretations and meanings (eg. Em_EI_IndInit refers to personal initiative as a determinant of integration)”

- 3rd sentence: what percent of datasets were double coded?

Double-coding was done for 20% of the state datasets. This is indicated on page 8.

Results

Individual experiences and meanings – collaboration and trust:

- 2nd paragraph/1st sentence: spell out what MSV stands for.

We had put in an acronym, but in order to protect confidentiality, are replacing this with

- 3rd paragraph: Use more quotes from respondents or state how the respondents’ perspectives are being relayed. This paragraph reads more like background than findings.

Agreed. We were telling, more than showing here. We have revised the paragraph to include a quote from respondents, then using that to elaborate a larger point, on pages 9-10: “Across states, we heard of individual practitioners exercising personal initiative to hasten improvements in infrastructure and service delivery. Following is an excerpt of an interview with an Ayurvedic doctor from a Delhi hospital: “there is a lack of storage space so the diagnosis room is being used for some storage. But I have been treating people in the Public Works Department and then it is getting resolved!” Many of the participants we spoke to in many states were familiar with each other – these personal relationships and interactions, more often than official platforms, were the basis for interaction, cross-referral, collective planning and advocacy, and in rarer cases, collaborative research.”

- 5th paragraph/3rd and 4th sentences: Elaborate on what is meant by “little something.”

We have gone on to show further what the participant meant by a little something – a demand for regular systematic, meetings. This is indicated on page 10, as follows:

As an Ayurvedic practitioner in Delhi put it, “if one takes a personal interest, there can be a little something. But everyone is busy in their own work. If it is done officially – like in a month, every 2nd Saturday ... Then it will happen more systematically.”

Group or system-linked experiences and meanings-distrust and fragmentation:

- 1st paragraph/last sentence: This sentence seems contrary to the section heading. Consider adding a section on differences observed across ages if you have enough data to do so.

Based on comments from Reviewer 1, we have flipped the section heading to talk about fragmentation first, and then distrust. We have indicated this drawing directly from the data on page 11:” More junior practitioners noted that even with respect to TCAM systems: “We three [Ayurveda, Unani, and Homeopathy] are together here, but cross-reference is very, very less...We don’t know what is the strong point of Ayurveda, Unani. Allopath will not know the strong point of homeopathy, Ayurveda. They just say ‘skin!’ – that’s all they know!”

- 3rd paragraph/2nd sentence: Rather than describing what you observed, use quotes or summaries of what the respondents said. For example “non-allopathic practitioners talked about the room assignments...” If you want to talk about your own observations, this should be added to the methods section as a data source and the methods of observation should be described.

This finding was based on both observations and remarks made by practitioners. We have included observations in our methods section, on page 8, cited them in the relevant section on page 13 (“We observed in many dispensaries and hospitals in Delhi that non-allopathic practitioners were assigned rooms on the top floor of the facility (Fieldnotes June 11th, 20th, 21st, 22nd, and 27th 2012”), and included direct quotations from participants to this effect:

Conclusion

The conclusion would be more salient if one or two examples were provided of an individually tailored strategy that would aid in integration.

This is a great idea! Based on comments from both Reviewer 1 and 2, we have proposed recommendations strategies and indicated them in Table 2.

General comment

The manuscript needs copy editing.

The manuscript has been carefully copy edited and revised to correct typographical errors, repeated words, and other mistakes.

VERSION 2 – REVIEW

REVIEWER	Eran Ben-Arye Lin Medical Center, Faculty of Medicine, Technion-Israel Institute of Technology, Israel
REVIEW RETURNED	06-Jun-2014

GENERAL COMMENTS	This is an important large-scale study performed in the field of integrative medicine.
--

REVIEWER	Barbara Stussman Barbara Stussman National Center for Complementary and Alternative Medicine National Institutes of Health Bethesda, Maryland USA
REVIEW RETURNED	11-Jun-2014

GENERAL COMMENTS	Methods 4th paragraph • Suggest adding “in-person” to end of sentence starting with “We would typically contact providers via cell phone...”
--

	Final paragraph  • The added explanations of a priori, emergent, and analytic codes are helpful. Suggest deleting the parenthesis with the code names and what they refer to. Results Table 2:  • The tables are a helpful addition to the findings. Table 2 is confusing because the plus signs refer to both promoting positive findings and mitigating negative findings. Suggest modifying column titles to reflect this (i.e. Strategies to increase facilitators/Strategies to decrease barriers). Authors can be trusted to review these minor suggestions and make changes. The manuscript needs proofreading for duplicate words, spacing, and fragments.
--	--

VERSION 2 – AUTHOR RESPONSE

-Editors Comments to Authors:

Please include a statement regarding ethics and any competing interests the authors may have.

We had included the following text on page 6 of our manuscript. “The research protocol was approved by the Institutional Ethics Committee of the Public Health Foundation of India.”

We have no competing interests; the declaration is made on Page 2 and in the submission form.

-Reviewer(s) Reports:

Reviewer: 1

Reviewer Name Eran Ben-Arye

Institution and Country Integrative Oncology Program, The Oncology Service and Lin Medical center, Clalit Health Services, Haifa and Western Galilee District, Israel; and Complementary and Traditional Medicine Unit, Department of Family Medicine, Faculty of Medicine, Technion-Israel Institute of Technology, Haifa, Israel

Please state any competing interests or state ‘None declared’: None declared

This is an important large-scale study performed in the field of integrative medicine. There is some discrepancy between the high standard of writing in the introduction and methods sections compared with the results section. I would recommend focusing the results section by presenting the findings on several theme axis (e.g. integrative vs. alternative conceptualization, patient-centered care vs. disease-oriented approach).

The need to present findings on theme axes is well taken. We have done this in the body of the manuscript and also provided a summary table for clarity.

Table 1. Summary of Findings

	Factors at Individual/Interpersonal Level		Factors at Group/System Level
Facilitators	A) Collegiality between practitioners within facilities B) Stature of TCA doctors C) Personal initiative of TCA doctors D) Personal experience of allopaths E) Political will of senior health system actors	Barriers	A) Fragmentation of jurisdiction and facilities B) Inter-system isolation and lack of communication C) Lack of trust and awareness of TCA systems D) Inadequate infrastructure and resources for TCA service delivery

In the discussion section, I recommend adding point-to-point recommendations for further integration based on each of the findings indicated in the results section. It is advisable to generalize your recommendations so that other scholars who promote integration worldwide will benefit from your experience.

Reviewer 2 asked for state-tailored recommendations, but we tend to agree with this reviewer that the lessons may be relevant in other country contexts. We have presented them in relation to our findings and further argued for their customization in each state.

Table 2. Recommendations to promote/address Integration, responding to findings

	Facilitators				Barriers			
	Collegiality	Stature	Personal Initiative	Personal Experience	Fragmentation	Isolation	Lack of trust/awareness	Inadequate infrastructure/
High level political will required for all strategies								
Case documentation and sharing across systems, and in the academic literature		+	+	+		+	+	

Routine opportunities for interaction and collaboration across systems (eg. health camps, health promotion drives)	+			+	+	+	+	+
Routine opportunities for interaction within co-located facilities (eg. staff meetings)	+			+	+	+	+	+
Rewards for integrative initiative of individuals (eg. challenge grants or institutional recognition)	+	+	+	+				
Rewards for integrative initiative at systems or facility level (eg. Joint targets like no of monthly referrals, no of cases jointly resolved)		+		+	+	+	+	+
Guidelines for collaboration (criteria and conditions for cross-referral, jointly developed by practitioners, non-clinical aspects of work together, including health promotion and managerial duties)				+	+	+	+	+

In addition, please add a small map indicating the 3 research areas for those readers who are not acquainted with the geography of India.

We have found an open-source map and indicated the three areas as Figure 1. We have reproduced this image below.

Figure 1. States in India where fieldwork was conducted

Reviewer: 2

Reviewer Name Barbara J. Stussman

Institution and Country Survey Statistician

Office of the Director

National Center for Complementary and Alternative Medicine

National Institutes of Health

U.S.A.

Please state any competing interests or state 'None declared': None declared

Abstract

Participants/2nd sentence:

Explain, "utilizing criteria of proximity from district headquarters" (i.e. did you use districts close by or did you vary the proximity?) If the former is true, this should be discussed in limitations.

Of the two administrative regions chosen within district, the nearest and farthest regions from district headquarters were selected for interviewing the participants. We have indicated this on page 7.

Methods

First paragraph: Is the larger research study published? If so, cite it. If not, provide a brief background on it.

The larger research study is reported, but not published. We have provided a brief background on page 6, as follows. "This analysis draws from a larger mixed methods implementation research study aimed at understanding operational and ethical challenges in integration of TCA providers for delivery of essential health services in three Indian states. The study looked at the contents and implementation of TCA provider integration policies in 3 states and at national level examining the understanding and interpretations of integration from the perspectives of different health systems actors. These coupled with their experiences in the actual processes of integration of TCA providers were studied using qualitative interview methods to help identify systemic and ethical challenges. Based on this, the study sought to derive strategies to augment the integration of TCA providers in the delivery of essential health services."

3rd paragraph:

- suggest providing number in parentheses after each category of respondent to show how many interviews were conducted with each.

This has been indicated as follows on pages 6-7: "Our methods comprised semi-structured, in-depth face-to-face interviews with policymakers (N=12), administrators (N=43), TCAM practitioners, (N=59) and allopathic practitioners (N=37), traditional healers (N=7), as well as health workers and community representatives (N=38) in three diverse Indian states: Kerala, where a number of systems have strong historical and systemic roots (N=74), Meghalaya, where local health traditions hold sway (N=61), and Delhi, where national, state, and municipal jurisdictions interface with multiple systems of medicine (N=61).

- State the length or range of length of each interview.

This has been indicated on page 7 of the paper, as follows: "Interviews, ranging from 20 to 90 minutes in length were undertaken, only with prior informed consent, and separate consent to record interviews."

- State mode of interviews (e.g., **face-to-face**, telephone).

The modality was face-to-face interviewing, which has been indicated on page 6 (please see quote above)

4th paragraph:

- State how respondents were approached and if you experienced hesitance in participating on the part of the respondent. If so, how did you counteract this hesitance?

This information has been added into Page 7 as follows: "We would typically contact providers via cell phone, share information about the study verbally or via email, and set up a time to interview them. In some cases, we would arrive during out-patient clinic hours to the chosen facility, share our participant information sheet and seek an appointment time with eligible participants. In most cases, we found that participants were keen to participate once they were aware of the nature of the study and, in some cases, the assurance of confidentiality. We had no refusals, although some allopathic practitioners had to be persuaded to participate by emphasizing that this study was not "pro-TCAM integration" per se, but merely seeking to understand state policy implementation."

- Provide the interviewing protocol as an appendix or describe it in the methods section.

The interviewing protocol is being submitted as an appendix.

5th paragraph:

- 2nd sentence: Provide information about the a priori codes and how they were developed.

A priori codes were derived directly from our research questions. This is indicated on page 8 as follows: "A priori codes were based on our research questions, reflecting experiences, interpretations and meanings of integration (eg. Tc_Ap_EI_Adm refers to a TCAM providers' explanation of experience of interactions with administration in the facility or the health care system). Emergent codes were used to describe the content or categories of these experiences, interpretations and meanings (eg. Em_EI_IndInit refers to personal initiative as a determinant of integration)"

- 3rd sentence: what percent of datasets were double coded?

Double-coding was done for 20% of the state datasets. This is indicated on page 8.

Results

Individual experiences and meanings – collaboration and trust:

- 2nd paragraph/1st sentence: spell out what MSV stands for.

We had put in an acronym, but in order to protect confidentiality, are replacing this with <Name of Well Known Ayurvedic Physician from Kerala>

- 3rd paragraph: Use more quotes from respondents or state how the respondents' perspectives are being relayed. This paragraph reads more like background than findings.
Agreed. We were telling, more than showing here. We have revised the paragraph to include a quote from respondents, then using that to elaborate a larger point, on pages 9-10: "Across states, we heard of individual practitioners exercising personal initiative to hasten improvements in infrastructure and service delivery. Following is an excerpt of an interview with an Ayurvedic doctor from a Delhi hospital: "there is a lack of storage space so the diagnosis room is being used for some storage. But I have been treating people in the Public Works Department and then it is getting resolved!" Many of the participants we spoke to in many states were familiar with each other – these personal relationships and interactions, more often than official platforms, were the basis for interaction, cross-referral, collective planning and advocacy, and in rarer cases, collaborative research."

- 5th paragraph/3rd and 4th sentences: Elaborate on what is meant by "little something."

We have gone on to show further what the participant meant by a little something – a demand for regular systematic, meetings. This is indicated on page 10, as follows:

As an Ayurvedic practitioner in Delhi put it, "if one takes a personal interest, there can be a little something. But everyone is busy in their own work. If it is done officially – like in a month, every 2nd Saturday ... Then it will happen more systematically."

Group or system-linked experiences and meanings-distrust and fragmentation:

- 1st paragraph/last sentence: This sentence seems contrary to the section heading. Consider adding a section on differences observed across ages if you have enough data to do so.

Based on comments from Reviewer 1, we have flipped the section heading to talk about fragmentation first, and then distrust. We have indicated this drawing directly from the data on page 11: "More junior practitioners noted that even with respect to TCAM systems: "We three [Ayurveda, Unani, and Homeopathy] are together here, but cross-reference is very, very less... We don't know what is the strong point of Ayurveda, Unani. Allopath will not know the strong point of homeopathy, Ayurveda. They just say 'skin!' – that's all they know!"

- 3rd paragraph/2nd sentence: Rather than describing what you observed, use quotes or summaries of what the respondents said. For example "non-allopathic practitioners talked about the room assignments..." If you want to talk about your own observations, this should be added to the methods section as a data source and the methods of observation should be described.

This finding was based on both observations and remarks made by practitioners. We have included observations in our methods section, on page 8, cited them in the relevant section on page 13 ("We observed in many dispensaries and hospitals in Delhi that non-allopathic practitioners were assigned

rooms on the top floor of the facility (Fieldnotes June 11th, 20th, 21st, 22nd, and 27th 2012)", and included direct quotations from participants to this effect:

Conclusion

The conclusion would be more salient if one or two examples were provided of an individually tailored strategy that would aid in integration.

This is a great idea! Based on comments from both Reviewer 1 and 2, we have proposed recommendations strategies and indicated them in Table 2.

General comment

The manuscript needs copy editing.

The manuscript has been carefully copy edited and revised to correct typographical errors, repeated words, and other mistakes.